# ALCAT1-Mediated Pathological Cardiolipin Remodeling and PLSCR3-Mediated Cardiolipin Transferring Contribute to LPS-Induced Myocardial Injury

**DOI:** 10.3390/biomedicines12092013

**Published:** 2024-09-03

**Authors:** Dong Han, Chenyang Wang, Xiaojing Feng, Li Hu, Beibei Wang, Xinyue Hu, Jing Wu

**Affiliations:** 1Department of Anesthesiology, Union Hospital, Tongji Medical College, Huazhong University of Science and Technology, Wuhan 430022, China; donghanxhyy@163.com (D.H.); 18186655047@163.com (C.W.); fxj1816427@163.com (X.F.); anesthesia_huli@163.com (L.H.); wbbstudent@163.com (B.W.); xinyuehu1117@163.com (X.H.); 2Institute of Anesthesia and Critical Care Medicine, Union Hospital, Tongji Medical College, Huazhong University of Science and Technology, Wuhan 430022, China; 3Key Laboratory of Anesthesiology and Resuscitation, Huazhong University of Science and Technology, Ministry of Education, Wuhan 430022, China; 4Department of Pain Management, Zhongnan Hospital of Wuhan University, Wuhan 430071, China; 5Department of Anesthesiology, Wuhan Fourth Hospital & Puai Hospital, Tongji Medical College, Huazhong University of Science and Technology, Wuhan 430030, China

**Keywords:** septic cardiomyopathy, cardiolipin, pathological CL remodeling, translocation of CL

## Abstract

Cardiolipin (CL), a critical phospholipid situated within the mitochondrial membrane, plays a significant role in modulating intramitochondrial processes, especially in the context of certain cardiac pathologies; however, the exact effects of alterations in cardiolipin on septic cardiomyopathy (SCM) are still debated and the underlying mechanisms remain incompletely understood. This study highlights a notable increase in the expressions of ALCAT1 and PLSCR3 during the advanced stage of lipopolysaccharide (LPS)-induced SCM. This up-regulation potential contribution to mitochondrial dysfunction and cellular apoptosis—as indicated by the augmented oxidative stress and cytochrome c (Cytc) release—coupled with reduced mitophagy, decreased levels of the antiapoptotic protein B-cell lymphoma-2 (Bcl-2) and lowered cell viability. Additionally, the timing of LPS-induced apoptosis coincides with the decline in both autophagy and mitophagy at the late stages, implying that these processes may serve as protective factors against LPS-induced SCM in HL-1 cells. Together, these findings reveal the mechanism of LPS-induced CL changes in the center of SCM, with a particular emphasis on the importance of pathological remodeling and translocation of CL to mitochondrial function and apoptosis. Additionally, it highlights the protective effect of mitophagy in the early stage of SCM. This study complements previous research on the mechanism of CL changes in mediating SCM. These findings enhance our understanding of the role of CL in cardiac pathology and provide a new direction for future research.

## 1. Introduction

Sepsis refers to critical organ dysfunction caused by an uncontrolled host response to infection, and the condition carries a high morbidity and mortality across the globe [1]. SCM represents a distinct form of cardiac dysfunction that occurs during sepsis and significantly impacts patient outcomes [2]. SCM is characterized by reversible cardiac dysfunction with increased cardiac apoptosis and irreversible myocardial injury [3]. Due to its intrinsic inability to regenerate, myocardium is incapable of self-rehabilitation after apoptosis. Moreover, the ongoing inflammatory response greatly exacerbates the reconstruction of the remaining non-apoptotic myocardium, thereby increasing the risk of myocardial infarction.

Growing evidence suggests that mitochondrial dysfunction is a critical factor in SCM [4,5]; this dysfunction has been linked to excessive oxidative stress and cardiomyocyte apoptosis [6]. Recent findings have shown that the lipids within mitochondrial membranes could modulate the membrane’s pro-apoptotic action, particularly through lipid-modifying events like phospholipid oxidation or lipid transfer [7]. CL is a crucial membrane phospholipid of mitochondria and is essential for maintaining the structural and functional integrity of mitochondria. CL, the signature phospholipid of mitochondria, presents almost exclusively in the inner mitochondrial membrane and constitutes up to 20% of mitochondrial lipids in mammals [8]. Altered forms of CL play a vital role in regulating mitochondrial-mediated cellular events.

ALCAT1 is a lysocardiolipin acyltransferase with a role in catalyzing the deleterious remodeling of CL [9]. Previous studies have demonstrated that ALCAT1-mediated pathological CL remodeling contributed to the mitochondrial dysfunction associated with various cardiac disorders, including hypertrophic cardiomyopathy, coronary heart disease, heart failure, and so on [10,11,12]. These aberrant CL species resulting from remodeling are believed to be more susceptible to oxidative damage [13]. It is speculated that this would have a negative impact on the interaction between cardiolipin and the electron transport chain complex, causing increased reactive oxygen species (ROS) generation and mitochondrial dysfunction [14]. The externalization of CL is considered an important step in triggering a series of cellular events, which encompass autophagy, mitochondrial fission and fusion, and cell death [15,16,17,18]. It has been reported that Phospholipid scramblase 3 (PLSCR3) exerts an essential role in CL translocation between the inner and outer mitochondrial membranes [15] and is involved in CL-mediated processes like mitophagy-mediated cell survival and Cytc release-mediated apoptosis [19,20]. Despite this knowledge, the precise involvement of ALCAT1 and PLSCR3 in regulating CL homeostasis and mitochondrial apoptotic pathways during apoptosis in SCM has not been fully understood.

Mitophagy is an integral component of the mitochondrial quality control (QC) mechanism, tasked with the repair or replacement of dysfunctional mitochondria through the targeted delivery of damaged mitochondria to lysosomal enzymes for degradation [21,22]. A disruption in this process results in a diminished clearance of damaged mitochondria, precipitating a substantial release of cytochrome c and other pro-apoptotic proteins, ultimately triggering cell death [23,24].

In this study, we observed the influences of LPS on the expression level of ALCAT1 and PLSCR3 and their correlation with autophagy and apoptosis in vitro. From the view of CL modifying, the present study will explore the mechanisms of CL remodeling and pathological CL transferring and their implications for autophagy and apoptosis. The findings will provide a novel concept and control strategy to interpret and handle sepsis and the related multiple organ dysfunctions.

## 2. Materials and Methods

### 2.1. Cell Culture

HL-1 cells were cultivated at a temperature of 37 °C in DMEM medium (GIBCO, Waltham, MA, USA) enriched with 10% fetal bovine serum (GIBCO, Burlington, ON, USA, Origin: Sydney, Australia), and fortified with 1% penicillin/streptomycin. The cells were maintained in a controlled environment with an atmosphere of air balanced with 5% CO_2_. HL-1 cells were purchased from Warner Bio Co., Ltd. (Wuhan, China). LPS was incorporated into the culture medium approximately 24 h post-seeding, once the cells had achieved a confluence level of 60–70%. The aim was to establish an in vitro model of LPS-induced septic cardiomyopathy.

### 2.2. Cell Viability

Cell viability was assessed using a 3-(4,5-dimethylthiazol-2-yl)-2,5-diphenyltetrazolium bromide (MTT) assay (Nanjing jiancheng, Nanjing, China). HL-1 cells were plated in a 96-well plate at a seeding density of 1 × 10^4^ cells per well. Subsequently, the cells were exposed to varying concentrations of LPS for different durations. Upon completion of the treatment period, media were aspired and replaced with 0.5 mg/mL MTT solution (20 μL/well), followed by a 4 h incubation at 37 °C. After removal of the supernatant, DMSO (150 μL/well) was then added to each well and shaken for 10 min on a shaker. By measuring the light absorption characteristics of MTT dye metabolites at 490 nm with an enzyme-labeled instrument (Thermo, Waltham, MA, USA), we can obtain a quantitative value related to the number of cells and metabolic activity, so as to evaluate the cell viability.

### 2.3. Measurement of Oxidative Stress

HL-1 cells were carefully seeded in a 6-well plate at a density of 2 × 10^5^ cells per well and subsequently underwent the designated experimental treatments. After that, cells were collected and lysed. The activity of superoxide dismutase (SOD) and the level of malondialdehyde (MDA) were determined following the manufacturers’ instructions. The aforementioned kits were all procured from the Jiancheng Bioengineering Institute (Nanjing, China).

### 2.4. Immunofluorescent Staining

HL-1 cells were plated in a 24-well plate containing gelatin-coated cover slips at a seeding density of 1 × 10^5^ cells per well and incubated with the corresponding treatment. Following the treatment, the cells on the slides were fixed using a PBS solution containing 4% paraformaldehyde for a duration of 30 min. Subsequently, the cells were blocked with 3% bovine serum albumin and then incubated with the primary antibody against LC3 proteins (Sigma, St. Louis, MO, USA) overnight at 4 °C, followed by three 5-min washes with PBS. Then, the slides were subsequently incubated with a secondary antibody conjugated with FITC (Life technologies, Paisley, UK) for 1 h. Finally, the samples were stained with 4′, 6-diamidino-2-phenylindole (DAPI; Sigma, St. Louis, MO, USA), a fluorescent nuclear stain, and observed using a confocal fluorescence microscope (Olympus, Tokyo, Japan). The green puncta indicated autophagosomes and the blue puncta indicated nuclei.

### 2.5. Electron Microscopy

After harvesting the HL-1 cells, they were fixed in 0.1 M phosphate buffer at pH 7.4—containing a mixture of 2% paraformaldehyde and 0.1% glutaraldehyde—for 1 h at room temperature. Following fixation, the cells underwent a dehydration process and were then embedded in 100% Eponate resin and left to harden. Ultrathin sections of the embedded cells were prepared and stained with uranyl acetate and lead citrate, and then viewed using an FEI Tecnai G2 12 model transmission electron microscope (FEI, Eindhoven, The Netherlands) equipped with a CCD camera.

### 2.6. Western Blot Analysis

The HL-1 cells were carefully plated into six-well plates and after treatment with experimental reagents according to the specified experimental designs, the cells were washed three times with PBS to eliminate any extraneous substances and lysed in lysis buffer to facilitate the extraction of cellular proteins for further analysis. After centrifugation at 12,000× *g* at 4 °C for 10 min, the supernatant containing the proteins was harvested. The protein concentrations of the cell extracts were quantified using a BCA Protein Assay Kit, ensuring accurate measurement for subsequent analysis. Equal amounts of protein from each sample were loaded on SDS–polyacrylamide gels for SDS-PAGE. The resolved proteins were transferred onto a membrane and subjected to standard immunoblot analysis to detect the presence of specific proteins. The intensity of the target protein bands was assessed was analyzed using Image-Pro Plus 6.0, which allowed for the calculation of the grey values, providing a quantitative measure of the protein expression levels. In the Western blot analysis, the following antibodies were utilized: P62/SQSTM1 (1:1000; RnD, catalog MAB8028), ALCAT1 (1:1000; Thermo, catalog PA5-25627), PLSCR3 (1:1000; Abcam, Cambridge, UK, catalog #ab137128), CytC (1:1000; Abcam, catalog #ab13575), BCL2, GAPDH (antgene, Wuhan, China), and HRP Goat anti-Rabbit igG (h+l) (antgene, Wuhan, China).

### 2.7. Statistical Analysis

Statistical analyses were conducted using GraphPad Prism 5.0 analysis software. All data are expressed as means ± standard error (SEM). A one-way analysis of variance (ANOVA) followed by Dunnett’s post hoc test was employed to assess the statistical significance of differences among mean values. Standardized tests, including assessments for normality and homogeneity of variance, were conducted. Normality was evaluated using the Shapiro–Wilk (SW) test, and the Kruskal–Wallis test, a nonparametric method, was applied when the data did not meet the assumption of normality. Homogeneity of variance was assessed using either the Brown–Forsythe test or Bartlett’s test. When the assumption of homogeneity of variance was not satisfied, a corrected one-way ANOVA was performed, which involved either the Brown–Forsythe or Welch’s ANOVA methods. The thresholds for statistical significance were set at * *p* < 0.05 and ** *p* < 0.01.

## 3. Results

### 3.1. The Effect of LPS on Cellular Injury and Oxidative Stress in the HL-1 Cell Line

In order to assess the cytotoxic impact of LPS on HL-1 cardiac cells, the cells were subjected to varying durations of LPS exposure (2, 4, 8, 16, and 24 h), followed by the implementation of the MTT assay. The outcomes, as illustrated in Figure 1a, reveal no noteworthy cell mortality in cells treated with 1 μg/mL of LPS at 2 and 4 h; however, the cell viability decreased significantly in 8–24 h. Additionally, we investigated the alteration of oxidative stress and measured the levels of oxidative stress markers, namely MDA and SOD. Figure 1b,c illustrates that there is no significant difference in the concentrations of SOD and MDA between the 4-h group and the control group; however, a noticeable decrease in SOD and an increase in MDA are observed in the 24-h group, indicating a significant increase in oxidative stress. Based on these findings, it is evident that LPS induces myocardial damage in a time-dependent manner. In this particular model, the severity of the damage correlates with different time points. Specifically, the 4-h and 24-h time intervals correspond to the early and late stages, respectively, in the development of SCM.

### 3.2. The Effect of LPS on Autophagy and Apoptosis in the HL-1 Cell Line

Autophagy and apoptosis are two cellular processes that have been extensively studied because they play an important role in the development of sepsis. In our study, we aimed to investigate the changes in autophagy markers, LC3II and P62, using Western blotting. The results revealed that the LC3-II protein level increased after LPS administration, reaching its peak at 4 h compared to the control group. Subsequently, the up-regulated levels of LC3-II gradually decreased over a 24-h period (Figure 2a). In addition, P62/SQSTM1 is a polyubiquitin-binding protein that is degraded during autophagy, and its down-regulation exhibited a significant time-dependent pattern (Figure 2b). Moreover, the observed changes in autophagy, as confirmed by fluorescence staining, were consistent with the protein expression of LC3-II revealed by Western blot analysis (Figure 2e). These findings suggest a time-dependent alteration in autophagy within septic cardiomyopathy, characterized by an initial increase followed by a gradual decline in autophagic activity.

Moving forward, we aimed to determine the levels of apoptosis-associated proteins at two different time points. HL-1 cardiomyocytes were treated with LPS for 4 and 24 h. When compared with the control group, the level of the antiapoptotic protein Bcl2 showed a significant decrease at 4 h, particularly at 24 h (Figure 2c). The release of the pro-apoptotic protein Cytc significantly increased at 4 h and continued to rise at 24 h (Figure 2d). These results indicate a significant shift in the cellular pattern toward a pro-apoptosis state in the later stage.

### 3.3. The Effect of LPS on Mitophagy in the HL-1 Cell Line

Mitophagy, a specialized form of cellular autophagy, is crucial in the pathogenesis of sepsis. To investigate the alterations in mitophagy levels, transmission electron microscopy (TEM) was employed to analyze HL-1 cells treated with 1 µg/mL of LPS for 4 and 24 h, allowing us to detect the formation of autophagosomes. Figure 3 illustrates the results, demonstrating that the control group displays a normal structure with appropriately distributed mitochondria. In contrast, exposure to LPS resulted in a significant increase in vacuoles surrounded by double membranes, indicating the presence of autophagosomes. Specifically, at the 4-h time point, we observed mitochondrial ridge structure disorder and even blank areas, as well as a higher number of enlarged autophagosomes containing mitochondria. Although the mitophagy level remained elevated at 24 h, it decreased compared to the 4-h time point, with continued severe mitochondrial damage. These changes in mitophagy coincided with those observed in macroautophagy, exhibiting an increase in early mitophagy and a decreasing trend in the late stages.

### 3.4. The Effect of LPS on CL Remodeling and CL Externalization in the HL-1 Cell Line

Previous studies have demonstrated that remodeling of CL may impact cell survival or apoptosis by modulating the process of mitochondrial autophagy. To explore the potential regulatory role of pathological CL remodeling in the regulation of sepsis-associated cardiomyopathy, the protein expression of ALCAT1 was assessed using Western blotting, and the corresponding results are presented in Figure 4a. Compared to the control group, the level of ALCAT1 gradually increased following the administration of LPS, with significant differences observed at 8 and 16 h. These findings depict the time-dependent expression pattern of the ALCAT1 protein in relation to the progression of septic cardiomyopathy. Consequently, this evidence suggests that ALCAT1-mediated pathological remodeling of CL during the later stages may play a crucial role in the development of septic cardiomyopathy.

The externalization of pathological CL is recognized as a crucial step in mediating autophagy dysregulation or cell apoptosis. To investigate the involvement of CL externalization, we performed Western blotting to assess the level of PLSCR3, a protein that mediates CL externalization. Figure 4b shows that the level of PLSCR3 protein greatly increases at 8 h and 16 h after LPS administration, compared to the control groups. These results were consistent with the changing pattern of the ALCAT1 protein, suggesting that CL externalization plays an important role in septic cardiomyopathy.

## 4. Discussion

This study has demonstrated that the expression of ALCAT1 and PLSCR3 increased significantly during the late stage of LPS-induced SCM. The up-regulation of ALCAT1 and PLSCR3 is believed to contribute to mitochondrial dysfunction and cell apoptosis, as evidenced by the increased oxidative stress and release of Cytc and the observed decreased mitophagy, as well as the decreased antiapoptotic protein Bcl2 and cell viability. Furthermore, LPS-induced apoptosis appeared at a later stage of increased autophagy and decreased mitophagy, suggesting that autophagy and mitophagy may be protective factors for HL-1 cells against LPS-induced SCM.

Autophagy and apoptosis are two critical cellular processes with distinct mechanisms, yet they exhibit crosstalk, displaying interaction or mutual inhibition under stress conditions, which is essential for regulating cellular fate. Previous research has shown that moderate up-regulation of autophagy is an adaptive response that protects organisms from a variety of insults [25]. In contrast, inadequate or excessive autophagy can be detrimental, leading to cell apoptosis [26]. In this study, we monitored the temporal progression of LPS-induced autophagy in HL-1 cells by evaluating the expression of LC3II and the degradation of P62. The results revealed a transient and significant induction of autophagy by LPS in the early stage. As the augmented autophagy declined in the late stage, LPS-induced cell apoptosis ensued, indicating the potential protective role of autophagy against LPS-induced cell injury. Furthermore, the expression of mitophagy, a critical mechanism for regulating mitochondrial quality, paralleled cellular autophagy, increasing in the early stage and decreasing in the late stage, potentially exerting a protective effect similar to cell autophagy.

Mitochondria-mediated cell apoptosis is an important pathway of programmed cell death. When damaged mitochondria are not effectively cleared and continue to be exposed to harmful stimuli, there is an increase in the mitochondrial membrane’s permeability, causing Cytc to be released from the mitochondria into the cytoplasm [27]. This subsequently activates a series of proteins that regulate apoptosis, ultimately inducing cell death [28]. In our study, we observed a reduction in protective mitophagy in the late stage of sepsis, which may result in severe mitochondrial damage and increased membrane permeability. Additionally, the up-regulation of Cytc release further validated our hypothesis. CL, as a characteristic phospholipid of mitochondria, is important in the process of mitochondria-mediated cell apoptosis. Therefore, we further investigated the potential role of CL in mediating cell apoptosis in our model.

Pathological remodeling of CL is a key contributor to the impairment of mitochondrial structure and function [10,12,29]. ALCAT1, a transmembrane receptor protein located on mitochondria-associated endoplasmic reticulum membranes, is abundant in the myocardium and liver and essential in CL remodeling [30]. Previous studies have demonstrated the role of ALCAT1 in catalyzing pathological remodeling of CL, its acyl composition was abnormal and found to be rich in DHA [11]. CL oxidation is a critical step in the process of CL remodeling, leading to diseases [14,31]. Under normal conditions, CL exhibits sensitivity to oxidative damage caused by ROS [11]. This sensitivity is attributed to its special location in the inner mitochondrial membrane, which is the primary site of ROS generation, as well as its high proportion of unsaturated fatty acids, particularly polyunsaturated fatty acids (PUFAs). In pathological states, ALCAT1 accelerates ROS production, resulting in increased generation of oxidized CL [12]. Oxidized CL not only leads to mitochondrial dysfunction but also becomes a new source of ROS, inducing ALCAT1 overexpression and further CL oxidation, forming a vicious cycle of self-destruction [9]. The elevated content of DHA in pathological CL exacerbates oxidative stress and promotes CL oxidation. ALCAT1-mediated CL pathological remodeling can accelerate reactive oxygen species (ROS) generation and CL oxidation, resulting in reduced CL content and consequent mitochondria-induced apoptosis in various conditions such as myocardial hypertrophy, heart failure, diabetes, aging, obesity, and other diseases [10,12,31,32]. The targeted inactivation of ALCAT1 prevents conditions like diet-induced obesity, non-alcoholic fatty liver disease (NAFLD), and MPTP-induced neurotoxicity, can ameliorate deficiency, and inhibits apoptosis [31,33,34]. In this study, we present findings on the time-dependent expression of the ALCAT1 protein in relation to the progression of SCM. The high expression of ALCAT1 is closely associated with increased oxidative stress and cell apoptosis, which aligns with the previous literature. Our results suggest that ALCAT1-mediated CL remodeling plays a significant role in the mitochondrial etiology of SCM.

In addition to the fact that abnormalities in CL content and composition have been linked to mitochondrial etiology of age-related diseases, CL transfer is believed to be closely related to the occurrence of downstream events as a key recognition signal [35]. For the time being, one of the PLS family members, PLSCR3, is located in the mitochondria and can regulate the CL externalization to the mitochondrial surface, thus participating in a series of cellular processes like apoptosis and mitophagy [19,20]. Previous studies showed that pro-mitophagy stimuli led to CL turnover from the inner to the outer mitochondrial membrane and enhanced mitochondrial mitophagy, as evidenced by the up-regulated protein expression of LC3-II and down-regulated P62 expression, which was dependent on the critical role of PLSCR3 [19]. Knockdown of PLSCR3 prevents stress-induced CL externalization [19,36]. While our study did not reveal a direct correlation between mitophagy and PLSCR3, as evidenced by the temporal expression pattern at different stages of SCM, the study did indicate that the expression time series of PLSCR3 is consistent with the up-regulation of ALCAT1, suggesting that PLSCR3 may be critical for the apoptotic process in the late stages of SCM. This observation may be linked to its involvement in mediating the process of aberrant CL translocation. 

Our research is centered on the study of SCM, a condition characterized by heart muscle dysfunction following sepsis. Notably, Takotsubo syndrome, also known as stress cardiomyopathy, exhibits similarities with SCM, particularly in its pathophysiological processes during sepsis. These similarities encompass a shared inflammatory cascade, oxidative stress, disrupted energy metabolism, and mitochondrial dysfunction [37,38,39,40]. Through the exploration of these common pathways, the findings from our investigation offer potential insights into the foundational mechanisms underlying Takotsubo syndrome. This suggests that the elucidated mechanisms could be broadly applicable and potentially contribute to the foundational research in understanding and managing Takotsubo syndrome.

There are several limitations in the current study. Firstly, this study is limited to cellular models, and the applicability of these findings to animal models requires further validation. Secondly, additional research is required to verify the production of pathological cardiolipin and its dynamic translocation across the mitochondrial membrane. Finally, while this study introduces a concept with significant potential, additional research is imperative to advance these findings toward clinical application as a potential intervention target.

## 5. Conclusions

In conclusion, our study has initially investigated the significant roles of ALCAT1 and PLSCR3 in SCM. These roles are related to the pathological remodeling of CL mediated by ALCAT1 and the translocation of pathological CL mediated by PLSCR3. Additionally, the worsening of myocardial injury is also associated with the reduction in mitophagy. This study provides a new insight into the CL-mediated mitochondrial mechanism of SCM. These findings enhance our understanding of the role of CL in cardiac pathology and provide a new direction for future research.

## Figures and Tables

**Figure 1 biomedicines-12-02013-f001:**
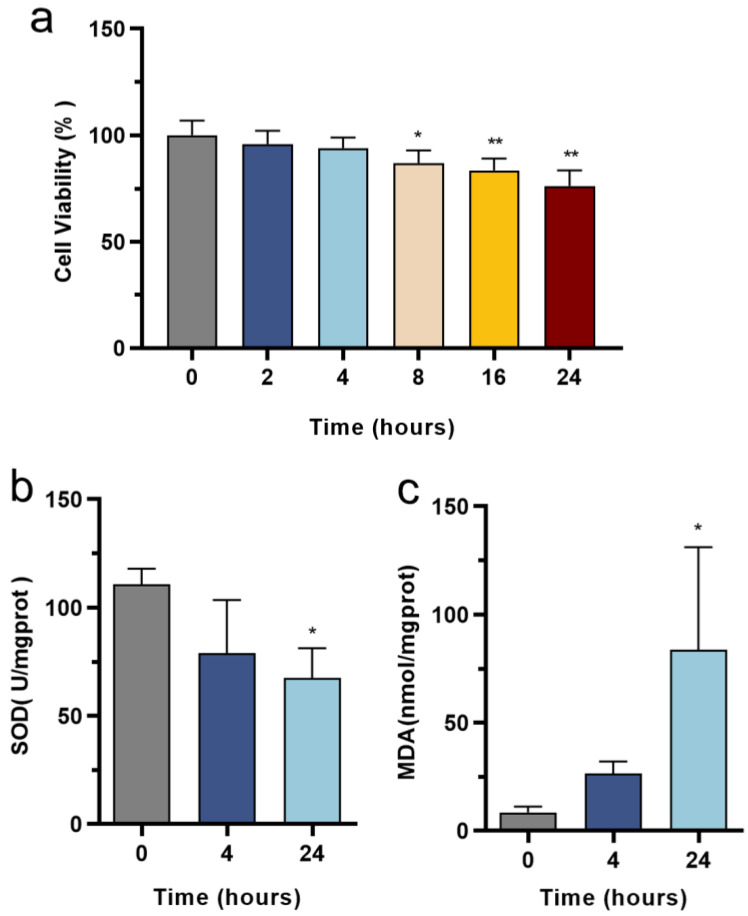
Changes in cell death and oxidative stress in the LPS-induced HL-1 cardiac cells: (**a**) cell viability was scored by methyl thiazolyl tetrazolium (MTT) assay, *n* = 5 independent experiments; (**b**) the level of SOD activity decreased in LPS-induced HL-1 cells, *n* = 3 independent experiments; (**c**) the level of MDA activity increased in LPS-induced HL-1 cells, *n* = 3 independent experiments. * *p* < 0.05, ** *p* < 0.01, compared with the control group.

**Figure 2 biomedicines-12-02013-f002:**
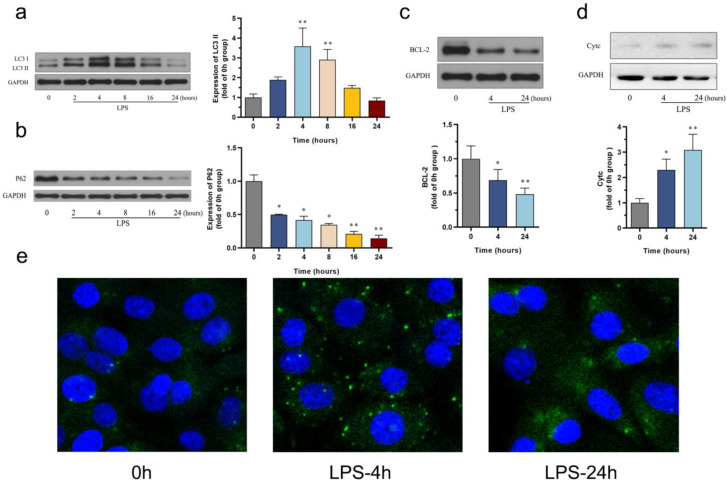
Changes in autophagy and apoptosis in the LPS-induced HL-1 cardiac cells: (**a**) Western blot images showing the time-course change in LC3-II in LPS-induced HL-1 cells, *n* = 5 independent experiments; (**b**) Western blot images showing the time-course change in P62 in LPS-induced HL-1 cells, *n* = 4 independent experiments; (**c**) Western blot images showing the change in BCL2 in LPS-induced HL-1 cells, *n* = 4 independent experiments; (**d**) Western blot images showing the change in cytochrome c in LPS-induced HL-1 cells, *n* = 3 independent experiments; (**e**) LC3 aggregation quantified under confocal fluorescence microscopy. * *p* < 0.05, ** *p* < 0.01, compared with the control group.

**Figure 3 biomedicines-12-02013-f003:**
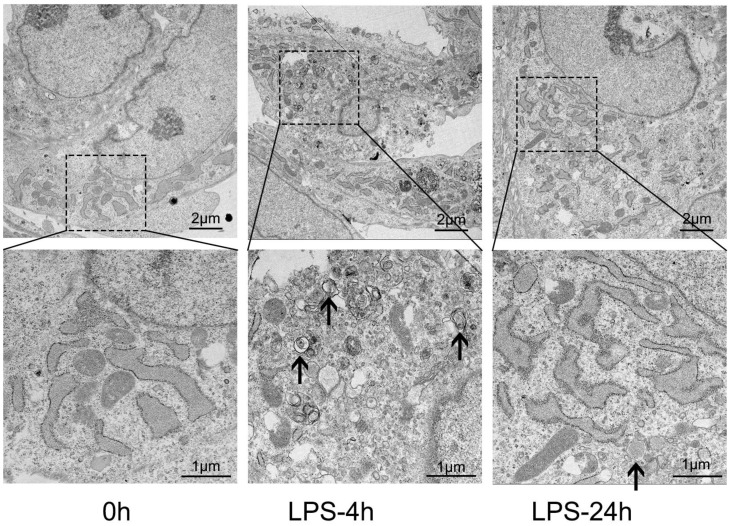
Changes in mitophagy in the LPS-induced HL-1 cardiac cells. Representative TEM images of HL-1 cells in control, LPS-treated for 4 h, LPS-treated for 24 h. The images below showcase an enlarged view delineated by the dashed boundary. Arrows, formation of autophagosomes.

**Figure 4 biomedicines-12-02013-f004:**
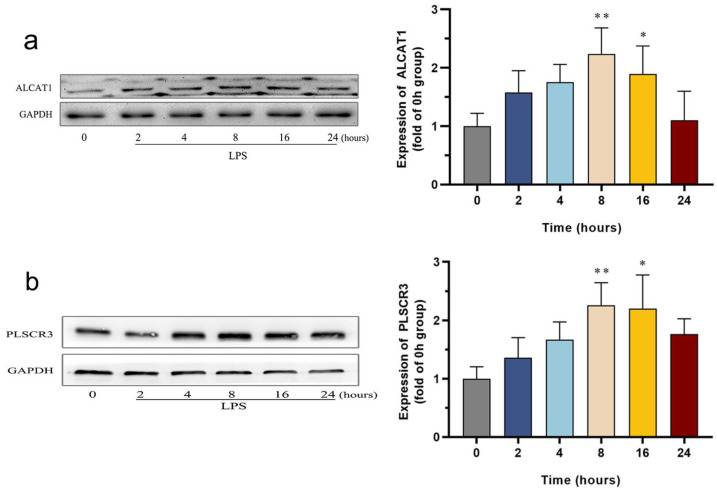
Changes in ALCAT1 protein and PLSCR3 protein in LPS-induced HL-1 cells: (**a**) Western blot images showing the time-course change in ALCAT1 in LPS-induced HL-1 cells, *n* = 4 independent experiments; (**b**) Western blot images showing the time-course change in PLSCR3 in LPS-induced HL-1 cells, *n* = 4 independent experiments. * *p* < 0.05, ** *p* < 0.01, compared with the control group.

## Data Availability

Data are contained within the article.

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
