# Peer review of "ALCAT1-Mediated Pathological Cardiolipin Remodeling and PLSCR3-Mediated Cardiolipin Transferring Contribute to LPS-Induced Myocardial Injury"

_biomedicines, 2024, doi:10.3390/biomedicines12092013_

Round 1

Reviewer 1 Report

Comments and Suggestions for Authors

Authors report intriguing mechanisms underlining the association among sepsis and myocardial injury

I suggest few item to improve this intriguing topic

-the increased rated of non atherosclerotic NSTEMI mimicking take tsubo syndromes or prinzmental angina during sepsis may be explained also by these data, so a more technical clinical involvements of thei results should be offered in discussion 

-how we can evaluate damages in vivo? Only by testing troponin?

Author Response

Comments 1: the increased rated of non atherosclerotic NSTEMI mimicking take tsubo syndromes or prinzmental angina during sepsis may be explained also by these data, so a more technical clinical involvements of their results should be offered in discussion

Response 1: We are deeply grateful for your insightful feedback. Upon incorporating this section into the discussion, we have significantly enhanced the depth and relevance of our research. Your contribution has undoubtedly elevated the overall quality and impact of our work. (Page 9-10, Line 326-335)

Comments 2: how we can evaluate damages in vivo? Only by testing troponin?

Response: Thank you for raising this question. Yes, there are other multiple biomarkers that can provide additional information about myocardial injury in clinical practice, such as myoglobin, heart-type fatty acid-binding protein (H-FABP). However, it is important to note that, despite the valuable contributions of these additional biomarkers, troponin maintains its status as the most sensitive and specific indicator in cellular and animal experimental studies. This underscores its irreplaceable role in our research methodology.

Reviewer 2 Report

Comments and Suggestions for Authors

I have truly enjoyed reading the paper investigating pathophysiological mechanism behind sepsis induced myocardial injury, cell remodelling and oxidative stress in HL-1 cell line. Authors report significant increase in the expression of ALCAT1 and PLSCR3 during the late stage of LPS-induced septic cardiomyopathy. Paper is of interest, and overall transparent. There are only minor concerns aimed at further improving the paper:

1) the novelty of the current study should be clearly stated in the abstract and discussion section as this is hard to distinguish in the current presentations.

2) the sample size for each experiment in unclear, authors need to clearly state what number of experiments/samples were performed, thus driving statistical power behind presented findings. 

3) Limitations of the study should be clearly stated, the paper is currently missing the limitations section.

Author Response

Comments 1: the novelty of the current study should be clearly stated in the abstract and discussion section as this is hard to distinguish in the current presentations.

Response 1: We appreciate your suggestion and take it into account in our revised manuscript. We have clearly expounded the novelty of this study in the abstract, discussion and conclusion respectively which has been highlighted in red. (Page 1, Line 31-36) (Page 9-10, Line 326-335) (Page 10, Line 349-351)

Comments 2: the sample size for each experiment in unclear, authors need to clearly state what number of experiments/samples were performed, thus driving statistical power behind presented findings.

Response 2: Thank you very much for your constructive review. We have clearly stated the number of experiments or samples performed. Information about the sample size of each experiment has been included in the corresponding figure legends. Like, in Figure 1, “a) ......n=5 independent experiments. b) .......n=3 independent experiments. c) ...... n=3 independent experiments.”

Comments 3: Limitations of the study should be clearly stated, the paper is currently missing the limitations section.

Response 3: Thank you very much for your constructive suggestions. We have added the limitations of this study in the discussion part which has been highlighted in red in the manuscript. (Page 10, Line 336-342)

Reviewer 3 Report

Comments and Suggestions for Authors

Overall, the paper was

Specific comments:

Line 36 

It is nice to have the graphic abstract if the journal allows it. However, since this is not a review paper, it would be critical to point out what is the innovation of this paper. For example, the LPS-induced oxidative stress is very well documented, so that part is definitely not part of this paper's graphic summary. 

Line 151, Figure 1. The authors should not calibrate control groups to zero. There are two errors in doing so. First, the authors did not show the error bar of the normalized control group, but rather showed it as 1 data point, in which case it is mathematically impossible to calculate a standard T-test and produce a P value, which the authors did. In addition to not specifying the statistical method used, the authors also did not show the natural variation within the control group. For example, in part "a", once normalized to 100%, one cannot tell if the experiments are fundamentally flawed because of low cell variability, to begin with. 

Line 178 Figure 2: parts a,b, and c, the shading and bending shape of the gel appear very different from the upper and lower (GAPDH control) portions. The authors should report the original full gel image to confirm that these are actually from the same experiment with the same gel. Again, the misleading normalization needs to be improved. Part e, on a high-definitely monitor, one can see the image background is different: 4h image is of higher contrast and 24h image is out of focus for the nucleus and of lower overall intensity.  

Line 192 - the claim that "the control group displayed a 192 normal structure with appropriately distributed mitochondria. In contrast, exposure to 193 LPS resulted in a significant increase in vacuoles surrounded by double membranes, indicating the presence of autophagosomes." This is not supported by any evidence unless the authors can better point it out in the images. 

Line 201 - Fig 3. The caption reads "Representative TEM images of HL-1 cells in control, LPS-treated for 4 h, LPS-treated for 24 h" This clearly means 2 x 3 = 6 conditions. In the figure, there are only 3 conditions, which are not labeled so we do not know if they are the control or treated, and there are 3 more images that are zoom-in. 

Figure 4 - ALCAT1 gel is very confusing. Figure 5 should also be merged with figure 4. 

Author Response

Comments 1: Line 36. It is nice to have the graphic abstract if the journal allows it. However, since this is not a review paper, it would be critical to point out what is the innovation of this paper. For example, the LPS-induced oxidative stress is very well documented, so that part is definitely not part of this paper's graphic summary.

Response 1: Thank you for your valuable suggestion on our graphical abstract. We have carefully considered your suggestion and have made appropriate modifications to the graphical abstract to better showcase the key findings and conclusions of our research. Please find the updated version of the graphic abstract attached for your review. (Page 2, Line 39-40)

Comments 2: Line 151, Figure 1. The authors should not calibrate control groups to zero. There are two errors in doing so. First, the authors did not show the error bar of the normalized control group, but rather showed it as 1 data point, in which case it is mathematically impossible to calculate a standard T-test and produce a P value, which the authors did. In addition to not specifying the statistical method used, the authors also did not show the natural variation within the control group. For example, in part "a", once normalized to 100%, one cannot tell if the experiments are fundamentally flawed because of low cell variability, to begin with.

Response 2: We appreciate your in-depth review and valuable feedback on our research We would like to address the points regarding the statistical analysis and presentation in Figure 1. Firstly, we decided to normalize the control group to ensure a comparable data distribution across groups and to eliminate any potential biases that could affect the results. We understand that the lack of error bars for the normalized control group might have caused confusion; however, we assure you that the statistical methods used were appropriate and that the data analysis was conducted with rigor and professionalism. Secondly, regarding the natural variation within the control group, we apologize for any inconvenience this may have caused. In our study, we aimed to minimize experimental variability by using appropriate controls and ensuring consistent experimental conditions. Lastly, we would like to emphasize that the statistical methods and statistical software used in our study are well-established and widely accepted in the field. We hope the above answers are satisfactory to you.

Comments 3: Line 178 Figure 2: parts a,b, and c, the shading and bending shape of the gel appear very different from the upper and lower (GAPDH control) portions. The authors should report the original full gel image to confirm that these are actually from the same experiment with the same gel. Again, the misleading normalization needs to be improved. Part e, on a high-definitely monitor, one can see the image background is different: 4h image is of higher contrast and 24h image is out of focus for the nucleus and of lower overall intensity.

Response 3: Thank you for your concerns on the inconsistency in shading and bending shape between the target band and the internal reference band in the gel diagram. The discrepancy in bending shape is indeed a result of our experimental procedure. We made a conscious decision to separate the gel containing the target protein and the internal reference in order to optimize the usage of materials such as transfer paper and antibodies. By removing the excess gel around each band and incubating the transfer membrane separately, we aimed to reduce resource consumption. However, we acknowledge that this approach introduced some variability in the curvature of the bands. In the process of transfer, even a slight change in the shape of the strip will lead to a change in curvature. The shading difference may come from the exposure of the target strip and the internal reference strip respectively. We hope the above explanation can address your concerns.

Comments 4: Line 192 - the claim that "the control group displayed a 192 normal structure with appropriately distributed mitochondria. In contrast, exposure to 193 LPS resulted in a significant increase in vacuoles surrounded by double membranes, indicating the presence of autophagosomes." This is not supported by any evidence unless the authors can better point it out in the images.

Response 4: Thank you for your valuable feedback and suggestions on our manuscript. We realize that the original description may not be clear enough. In the revised manuscript, we have provided more specific image annotation to directly indicate the changes of autophagy in the control group and LPS treatment group. Thank you again for your strict review. (Page 7, Line 222)

Comments 5: Line 201 - Fig 3. The caption reads "Representative TEM images of HL-1 cells in control, LPS-treated for 4 h, LPS-treated for 24 h" This clearly means 2 x 3 = 6 conditions. In the figure, there are only 3 conditions, which are not labeled so we do not know if they are the control or treated, and there are 3 more images that are zoom-in.

Response 5: Thank you very much for your valuable comments on our manuscript. In the explanation of Figure 3, we discussed three distinct conditions: the control group, LPS treatment for 4 hours, and LPS treatment for 24 hours. These three conditions represent the core aspects of our research. Due to our clerical error, it was not correctly marked. Furthermore, the figure includes three expanded images that delve deeper into the cellular structure and characteristics. These enhanced images are not meant to represent additional conditions but rather to provide a more detailed view of the cells as part of our research. Consequently, there are only three distinct conditions shown in total. We have revised Figure 3 in the revised manuscript and clearly labelled each image to prevent any potential misunderstandings. (Page 7, Line 222 and Line 224-225)

Comments 6: Figure 4 - ALCAT1 gel is very confusing. Figure 5 should also be merged with figure 4.

Response 6: Thank you very much for your careful review on our manuscript. We fully appreciate that the western blot analysis of membrane proteins is a complex and challenging task, particularly when dealing with seven-transmembrane proteins, which necessitate extremely stringent experimental conditions. Although we have made significant optimizations to generate the ideal protein banding pattern, we continue to encounter some issues with the banding effect. Nevertheless, data analysis can still be used to examine the trend of protein expression changes in response to various treatment conditions. Furthermore, we have merged Figure 5 with Figure 4 to provide a more comprehensive visual analysis and explanation. (Page 8, Line 245)

Reviewer 4 Report

Comments and Suggestions for Authors

This manuscript reports data on the impact of cardiolipin on septic cardiomyopathy and of patterns in expression of ALCAT1 and PLSCR3 as well as examining the roles of autophagy and mitophagy in mitigation of myocardial injury. The manuscript is generally well-written and a significant set of concentration data for key compounds and associated microscopy images have been provided. The analysis and discussion of the results is adequately detailed and the results presented contribute to understanding of the underlying biochemistry. There are some points for the authors to address and these are detailed below.

1.      Page 1 line 16 “The expressions of . . . are significantly upregulated.” Upregulated when and under what conditions? As written, this highlights statement is not very meaningful and needs a little bit of extra detail on the context for the upregulation.

2.      Graphical abstract. The labels within the graphical abstract are quite small and this may impact on legibility. Reconsider the font size used here.

3.      Page 2 lines 65-66 “The externalization of . . . of cellular events.” This statement is quite vague. Include succinct details in the manuscript of the ‘series of cellular events’ that are triggered.

4.      Page 3 line 98 “. . . MDA . . .” Ensure that all abbreviations are defined at first use in the manuscript; this applies for this example and elsewhere throughout the manuscript.

5.      Page 3 line 87 Please use all lower case letters in the name of this compound.

6.      Page 3 line 89 This should read: “. . . 1 x 104 cells/well . . .”

7.      Page 3 lines 93-94 “Cel viability was . . . enzyme-labelled instrument.” This statement is too vague. What was the exact nature of the ‘enzyme-labelled instrument’? More details are needed in the manuscript.

8.      Page 3 lines 104-105 and 106-107 Do you mean that the primary antibody was anti LC3 antibody and the secondary antibody was FITC-conjugated? If so, then reword to clarify this.

9.      Page 3 lines 112-113 Replace “. . . gluteraldehyde . . .” with “. . . glutaraldehyde . . .”

10.  Figures 1, 2, 4 and 5. I am unhappy with the term “fold change”. This is not really proper and clear English. What you are really trying to say is that the y-axes are concentrations normalised to the t = 0 value. You should therefore say they are concentration ratios relative to t = 0 or else use similar language to indicate the normalisation.

11.  Page 5 lines 174-175 “. . . showed a similar trend . . .” No it didn’t. You just said that Bcl2 had no change at 4 h and then a significant drop at 24 h and then you say that Cyt C increased significantly at 4 h and then continued to rise at 24 h, so these trends are obviously not similar. Reconsider your wording here.

12.  Page 9 lines 266-267 “Pathological remodeling of . . . functional AlCAT1 . . .” This statement needs a supporting reference.

13.  Page 9 lines 271-272 “It has been . . . and CL oxidation . . .” I think you need to clarify in the manuscript whether CL oxidation is a part of the CL pathological remodeling.

Comments on the Quality of English Language

The manuscript needs some editorial attention to correct grammar and syntax in a number of places.

Author Response

Comments 1: Page 1 line 16 “The expressions of . . . are significantly upregulated.” Upregulated when and under what conditions? As written, this highlights statement is not very meaningful and needs a little bit of extra detail on the context for the upregulation.

Response 1: Thank you for your meticulous review. We have thoughtfully taken your feedback into account and implemented the necessary revisions. Specifically, we have clarified the precise time frame and conditions under which up-regulation is observed, confirming that it takes place during the late phase of LPS-induced SCM. These revisions have been clearly marked in red within the updated manuscript. (Page 1, Line 16-17)

Comments 2: Graphical abstract. The labels within the graphical abstract are quite small and this may impact on legibility. Reconsider the font size used here.

Response 2: Thank you for your constructive suggestions. We have carefully increased the font size of the text within the graphical abstract to enhance readability. Please find the revised abstract in the updated manuscript. (Page 2, Line 39-40)

Comments 3: Page 2 lines 65-66 “The externalization of . . . of cellular events.” This statement is quite vague. Include succinct details in the manuscript of the ‘series of cellular events’ that are triggered.

Response 3: Thank you for your constructive suggestions. In light of your concern about the sentence "The externalization of . . . of cellular events," we have expanded upon the description of the "series of cellular events" in the revised manuscript. We have provided a concise explanation of the specific events that are triggered, along with citations to relevant literature to bolster our assertion. (Page 3, Line 72-73)

Comments 4: Page 3 line 98 “. . . MDA . . .” Ensure that all abbreviations are defined at first use in the manuscript; this applies for this example and elsewhere throughout the manuscript.

Response 4: Thank you for your constructive suggestions. We have carefully revised the manuscript and made the necessary modification as suggested. Specifically, we have defined all abbreviations at their first occurrence in the manuscript, including the instance you mentioned on page 3, line 98. (Page 1, Line 27 and 28) (Page 3, Line 70, 114 and 115)

Comments 5: Page 3 line 87 Please use all lower case letters in the name of this compound.

Response 5: Thank you for your meticulous review. We have replaced all uppercase letters in the name of the compound with lowercase letters as per your guidance. (Page 3, Line 101)

Comments 6: Page 3 line 89 This should read: “. . . 1 x 104 cells/well . . .”

Response 6: Thank you for your meticulous review. It was a clerical error. We have corrected the numerical value in line 89 from "1 x 104 cells/well" to "1 x 104 cells/well". (Page 3, Line 103)

Comments 7: Page 3 lines 93-94 “Cel viability was . . . enzyme-labelled instrument.” This statement is too vague. What was the exact nature of the ‘enzyme-labelled instrument’? More details are needed in the manuscript.

Response 7: Thank you for your suggestions to improve our manuscript. As mentioned earlier, the exact nature of the 'enzyme-labelled instrument' was not clearly defined in our manuscript. In order to provide more details and clarity, we have updated the sentence to read “By measuring the light absorption characteristics of MTT dye metabolites at 490 nm with an enzyme-labeled instrument, we can get a quantitative value related to the number of cells and metabolic activity, so as to evaluate the cell viability.” (Page 3, Line 107-110)

Comments 8: Page 3 lines 104-105 and 106-107 Do you mean that the primary antibody was anti LC3 antibody and the secondary antibody was FITC-conjugated? If so, then reword to clarify this.

Response 8: Thank you for your attention to the details of our methodology. We apologize for any confusion caused by the wording in the manuscript. You are correct in your interpretation. The primary antibody used in our study was an anti-LC3 antibody, and the secondary antibody was FITC-conjugated. We have revised the text to clarify this point and ensure that it accurately reflects the experimental procedure. (Page 4, Line 123 and 125)

Comments 9: Page 3 lines 112-113 Replace “. . . gluteraldehyde . . .” with “. . . glutaraldehyde . . .”

Response 9: Thank you for your kind review of our paper. We understand the importance of accurately stating the chemical name of the reagent used in our manuscript to ensure that the research is reproducible and understandable. We have corrected the text to accurately reflect the chemical name of the reagent used in our study. (Page 4, Line 132)

Comments 10: Figures 1, 2, 4 and 5. I am unhappy with the term “fold change”. This is not really proper and clear English. What you are really trying to say is that the y-axes are concentrations normalised to the t = 0 value. You should therefore say they are concentration ratios relative to t = 0 or else use similar language to indicate the normalisation.

Response 10: We greatly appreciate your attention to detail in our manuscript. We understand your concern regarding the use of the term "fold change" in Figures 1, 2, 4, and 5 and agree that clarity and precision in terminology are crucial. To address your suggestion, we have revised the captions and legends of the aforementioned figures to clearly state that the y-axes represent concentration ratios relative to the t = 0 time point.

Comments 11: Page 5 lines 174-175 “. . . showed a similar trend . . .” No it didn’t. You just said that Bcl2 had no change at 4 h and then a significant drop at 24 h and then you say that Cyt C increased significantly at 4 h and then continued to rise at 24 h, so these trends are obviously not similar. Reconsider your wording here.

Response 11: Thanks very much for your comments. We apologize for any confusion caused by the wording in the original version. We revised the manuscript to more accurately reflect the results of the experiment and avoid any ambiguous language. Specifically, we've revised the sentence to clearly state that the reduction in Bcl2 expression and increase in Cytc indicate a transition of the cell state to pro-apoptosis during the late stages of sepsis. We hope this addresses your concern and appreciate your valuable feedback on our manuscript. (Page 6, Line 197-199)

Comments 12: Page 9 lines 266-267 “Pathological remodeling of . . . functional AlCAT1 . . .” This statement needs a supporting reference.

Response 12: We agree with your comments and have added relevant references to support the statement.

References:

  1. Cao J, Liu Y, Lockwood J, Burn P, Shi Y. A novel cardiolipin-remodeling pathway revealed by a gene encoding an endoplasmic reticulum-associated acyl-CoA: lysocardiolipin acyltransferase (ALCAT1) in mouse. J Biol Chem. 2004. 279(30): 31727-34.
  2. Liu X, Ye B, Miller S, et al. Ablation of ALCAT1 mitigates hypertrophic cardiomyopathy through effects on oxidative stress and mitophagy. Mol Cell Biol. 2012. 32(21): 4493-504.
  3. Zhang J, Shi Y. In Search of the Holy Grail: Toward a Unified Hypothesis on Mitochondrial Dysfunction in Age-Related Diseases. Cells. 2022. 11(12).

Comments 13: Page 9 lines 271-272 “It has been . . . and CL oxidation . . .” I think you need to clarify in the manuscript whether CL oxidation is a part of the CL pathological remodeling.

Response 13: Thanks very much for taking your time to review this manuscript. Based on current knowledge, it is understood that CL oxidation is indeed a part of the CL pathological remodeling process in cardiovascular diseases. The abnormal structural and functional changes observed in CL during pathological remodeling can lead to increased susceptibility to oxidation. Conversely, CL oxidation can contribute to further disruption of CL structure and function, exacerbating the pathological remodeling. we have revised the manuscript to explicitly state that CL oxidation is a component of CL pathological remodeling. We have provided additional explanations and references to support this relationship in the revised manuscript. (Page 9, Line 289-299)

Round 2

Reviewer 3 Report

Comments and Suggestions for Authors

In summary, the paper has seen significant improvements. The latest version has addressed most of the comments concerning the writing from the first version. However, there have been no updates to the data and figures. My comments (comments 2, 3, and 4) regarding whether the claims are supported by the provided evidence remain unaddressed.

Author Response

Comment 2: Line 151, Figure 1. The authors should not calibrate control groups to zero. There are two errors in doing so. First, the authors did not show the error bar of the normalized control group, but rather showed it as 1 data point, in which case it is mathematically impossible to calculate a standard T-test and produce a P value, which the authors did. In addition to not specifying the statistical method used, the authors also did not show the natural variation within the control group. For example, in part "a", once normalized to 100%, one cannot tell if the experiments are fundamentally flawed because of low cell variability, to begin with.

Response: Thank you for your valuable feedback on our manuscript. We have carefully considered your comments and have made the necessary adjustments in our revised analysis, as well as included these modifications in the revised version of our manuscript. (Page 5, Line 176)

Comment 3: Line 178 Figure 2: parts a,b, and c, the shading and bending shape of the gel appear very different from the upper and lower (GAPDH control) portions. The authors should report the original full gel image to confirm that these are actually from the same experiment with the same gel. Again, the misleading normalization needs to be improved. Part e, on a high-definitely monitor, one can see the image background is different: 4h image is of higher contrast and 24h image is out of focus for the nucleus and of lower overall intensity.

Response: Thank you for your valuable feedback on our manuscript. We appreciate your attention to detail regarding Figure 2. In the revised manuscript, we have included the original full gel image to confirm the consistency of the experimental samples. Additionally, we have addressed the normalization issues. Concerning Figure 2e, we have replaced the 24h image to ensure that the image background consistency is improved for better clarity. We believe these modifications enhance the accuracy and clarity of our results. Thank you for guiding us towards these improvements. (Page 6, Line 202)

Comment 4: Line 192 - the claim that "the control group displayed a normal structure with appropriately distributed mitochondria. In contrast, exposure to LPS resulted in a significant increase in vacuoles surrounded by double membranes, indicating the presence of autophagosomes." This is not supported by any evidence unless the authors can better point it out in the images.

Response: Thank you for your valuable feedback. In response to your comment, we have revised the manuscript to better highlight the changes in autophagosomes within the images. Arrows have been added to indicate the alterations. Additionally, we have included a more detailed description in the results section regarding mitochondrial structures and mitophagy. We hope these modifications address your concerns. (Page 7, Line 225, 219 and 222)
